

# Intelligent accounting optimization method based on meta-heuristic algorithm and CNN

Yanrui Dong

School of Accounting, Zhengzhou Vocational College of Finance and Taxation, Zhengzhou, Henan, China

## ABSTRACT

The evolution of social intelligence has led to the adoption of intelligent accounting practices in enterprises. To enhance the efficiency of enterprise accounting operations and improve the capabilities of accountants, we propose an intelligent accounting optimization approach that integrates meta-heuristic algorithms with convolutional neural networks (CNN). First, we enhance the CNN framework by incorporating document and voucher information into accounting audits, creating a multi-modal feature extraction mechanism. Utilizing these multi-modal accounting features, we then introduce a method for assessing accounting quality, which objectively evaluates financial performance. Finally, we propose an optimization technique based on meta-heuristic principles, combining genetic algorithms with annealing models to improve the accounting system. Experimental results validate our approach, demonstrating an accuracy of 0.943 and a mean average precision (mAP) score of 0.812. This method provides technological support for refining accounting audit mechanisms.

# INTRODUCTION

In the economy's dynamic evolution, the complexity of businesses continues to grow. Globalization exposes enterprises to expanded cross-border opportunities, necessitating adaptation to diverse regional accounting standards. Concurrently, evolving regulations impose new demands on financial reporting. Advances in information technology further enhance the efficiency of financial data processing and reporting. In this context, studying and continually optimizing accounting methods becomes crucial for financial transactions to accommodate larger and more intricate business environments. This effort contributes to the formulation of authoritative financial standards and enhances the comparability and transparency of corporate financial reporting (*Wang et al., 2022a*; *Zhou, 2022*).

Accounting constitutes an essential facet of business operations. Through deep exploration of accounting methods, more precise and credible financial reporting practices can be identified, bolstering the accuracy and credibility of financial data. Robust accounting practices facilitate the provision of comprehensive and transparent financial information, thereby serving as a foundation for intelligent decision-making by enterprise management and investors, thereby promoting efficient economic operations (*Hu, 2022*). Moreover, a

Corresponding author
Yanrui Dong,
Yanrui19937896822@126.com

thorough examination of accounting methods aids in establishing consistent international financial standards, fostering global financial information coordination, and enhancing global economic system stability. Amidst an ever-evolving economic landscape, an in-depth study of accounting methods enables enterprises to adapt and enhance their competitiveness effectively (*Chen & Chao, 2022*; *Cui & Verma, 2022*). Overall, the study of accounting methods enhances financial data precision within enterprises and contributes to the broader development of the financial sector within economic and regulatory frameworks.

Studying accounting methods involves numerous difficulties and challenges, which increase the complexity researchers face when proposing new methods or improving existing ones (*Berdiyeva, Islam & Saeedi, 2021*). Firstly, enterprises' business environment is increasingly complex, with diversified business models and transaction structures. Accounting methods must, therefore, adapt flexibly to this diversity and ensure that research outcomes are applicable across different industries and types of enterprises (*Mancini, Lombardi & Tavana, 2021*; *Zhuo, 2022*; *Yuan, 2023*). Secondly, regulations and accounting standards in various countries and regions are continually evolving. Researchers must stay updated with these changes to ensure that proposed accounting methods align with the latest regulatory requirements and maintain compliance. Lastly, in the era of big data, the vast and diverse datasets generated by enterprises pose a significant challenge in processing and analyzing data to extract useful financial information (*Wu, 2021*; *Li Cui, 2022*).

To overcome these challenges, researchers integrate expertise in accounting and adeptly apply advanced machine learning techniques. For instance, *Hernes & Sobieska-Karpińska (2016)* developed a consensus-based group accounting support system evaluated using data from the Warsaw Stock Exchange. *Qu (2023)* structured an intelligent accounting system into acquisition, storage, and analysis layers, enhancing decision-making capabilities through intelligent accounting evaluation systems. *Wang, Zhang & Tao (2021)* proposed an intelligent financial sharing platform based on convolutional neural network (CNN) to improve accounting efficiency, minimize costs, and enhance compliance. *Chen & Metawa (2020)* introduced a cloud-based enterprise financial management system that processes accounting data. *Desyatnyuk, Muravskyi & Shevchuk (2021)* innovated with unmanned and expert accounting models based on artificial intelligence (AI) theories. *Alazzazbi, Mustafa & Karage (2023)* constructed a financial system integrating accounting and risk management, while *Guo (2021)* utilized data mining to design a comprehensive financial analysis system. *Lai (2022)* explored an intelligent financial system framework and introduced digital mining-based accounting support. *Ding (2022)* developed an intelligent financial management model using digital technology. These innovative approaches leverage advanced technologies to address the complexities and demands of modern accounting, aiming to enhance financial management practices and support decision-making in enterprise contexts.

However, with the advancement of the times and the deepening of globalization, the business environment is becoming increasingly complex. In this rapidly changing era, the increasingly diversified business models and transaction structures of enterprises have made traditional accounting methods accurately capture and reflect the true financial

status of enterprises. In a rapidly changing business environment, enterprises need not just simple financial data recording but also a financial system that can reflect the operational status of the enterprise in a timely and accurate manner and provide strong support for decision-making. This article proposes an innovative intelligent accounting optimization method to address this challenge. This method is based on the powerful capabilities of meta-heuristic algorithms and CNN. Meta-heuristic algorithms possess the characteristics of global search and optimization, enabling them to find the optimal solution to a problem quickly.

Meanwhile, with its powerful image processing capabilities, CNN can accurately capture and identify complex financial data patterns. Combining the two enables this optimization method to provide enterprises with accurate financial accounting and forecasting based on a deep understanding of the complex business environment. By harnessing the capabilities of artificial intelligence and advanced algorithms, our intelligent accounting method aims to deliver more precise and reliable financial information. This will empower management to make informed decisions, enhancing enterprise competitiveness in today's competitive business landscape.

## RELATED WORKS

The intelligent accounting optimization method, integrating metaheuristic algorithms and CNN, combines the search and parameter optimization capabilities of metaheuristic algorithms with the feature extraction prowess of CNN to enhance intelligent accounting processes. This approach enables identifying optimal or near-optimal solutions within extensive search spaces, enhancing the system's ability to handle complex accounting problems autonomously. Automating decision-making processes reduces intervention requirements and increases accounting accuracy. This method aims to imbue accounting systems with greater intelligence, adaptability, and efficiency, providing enterprises with more precise and effective financial decision-making capabilities.

In recent years, various optimization methods in intelligent accounting have significantly enhanced the efficiency of enterprise financial management. *Kim et al. (2023)* combined cluster analysis with Data Envelopment Analysis (DEA) to analyze core competitiveness positioning and optimise accounting practices. *Su & Sun (2023)* evaluated enterprise performance based on accounting data, offering critical insights for investors, senior managers, and government decision-makers. *Bose, Dey & Bhattacharjee (2023)* addressed profit estimation inaccuracies by designing performance evaluation indicators linked to accounting data, reducing operational profit errors. *Li, Wang & Yang (2023)* focused on improving enterprise management and performance systems by analyzing traditional accounting method issues and proposing structural enhancements using BP neural networks. *Brusov & Filatova (2023)* applied linear analysis methods to explore accounting system impacts, while *Chang, Lee & Lee (2023)* conducted correlation analyses using profitability and relative value indicators alongside corresponding data. *Oyedele et al. (2023)* utilized financial indicators across profitability, operational capacity, cash flow, debt-paying ability, and growth potential dimensions to construct a performance evaluation

model, employing DEA for comprehensive ranking calculations based on listed company data.

Metaheuristics are algorithms that emulate the optimization processes observed in natural or social systems. These algorithms generally operate independent of specific problem structures, relying instead on heuristic rules inspired by natural selection, physics, or analogous phenomena. Their primary objective is to discover optimal or near-optimal solutions within a problem's solution space, iteratively refining candidate solutions. *Fontes, Homayouni & Goncalves (2023)* utilized the simulated annealing algorithm as a local search method and employed particle swarm optimization to tackle integrated production and transportation job scheduling under vehicle constraints. *Defersha, Obimuyiwa & Yimer (2022)* formulated a mathematical framework for flexible job resource scheduling, considering order-dependent constraints and integrating the simulated annealing algorithm. *Wang et al. (2022b)* addressed the scheduling of distributed pipelines with sequence-dependent constraints, proposing a two-stage iterative greedy approach. *Sun et al. (2023)* combined genetic algorithms and variable neighbourhood search to optimize scheduling, emphasizing load balancing through customized crossover and mutation operators. *Xu & Yu (2022)* introduced a dynamic shrinkage method using adaptive trade-off and evolution strategies to enhance efficiency and meet load requirements. *Abido & Elazouni (2021)* developed a multi-objective evolutionary approach for project scheduling involving multi-mode activities to minimize time and capital costs. *Pan et al. (2021)* presented a deep reinforcement learning-based optimization method for scheduling problems, employing convolutional neural networks to generate end-to-end feasible solutions trained with actor-critic methods and refined *via* metaheuristics. *Song et al. (2022)* utilized graph neural networks to learn priority scheduling rules, leveraging a heterogeneous architecture for offline training and rule relationship capture. *Tian, Liu & Lv (2024)* proposed a new nature-inspired meta-heuristic algorithm, the Snow Goose Algorithm, which mimics the flight patterns observed during their migratory process. *Jia, Lu & Xing (2024)* introduced an evolutionary updating mechanism called the Memory Backtracking Strategy, which comprises the thinking, recall, and memory phases. *Liang et al. (2024)* made several adjustments to the original Ant Colony Optimization algorithm to make it effective for the problem of maximizing influence.

## INTELLIGENT ACCOUNTING OPTIMIZATION METHOD BASED ON META-HEURISTIC AND CNN

To enhance corporate accounting operations and provide enterprises with high-quality financial data and decision-making advice, we propose an advanced method using meta-heuristics and convolutional neural networks (CNNs). This approach aims to analyze enterprise accounting intelligently, encompassing multi-modal accounting feature extraction (MAFE) using CNN, accounting quality assessment (AQAA), and meta-heuristics-based accounting optimization (MAO).

Figure 1 illustrates our technology roadmap. First, we introduce a CNN-based MAFE method to address accounting data's complex, multi-modal nature. Traditional methods

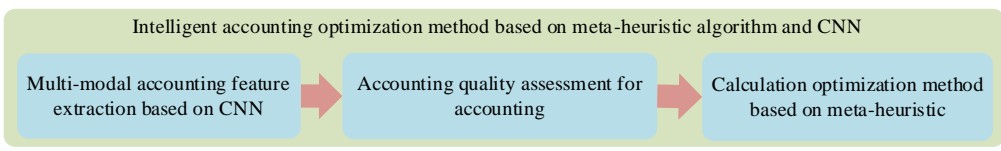

**Figure 1** **The technology roadmap of our method.**

often struggle to extract features from diverse financial and non-financial information comprehensively. CNN's robust feature learning capability enables the extraction of key features from numerical data and unstructured sources like text and images, providing critical support for subsequent AQAA and method optimization.

Next, we propose an AQAA method to evaluate the quality of extracted accounting features. This involves constructing a machine learning-based evaluation model to assess feature accuracy and reliability objectively. By identifying accounting issues promptly, this method guides the optimization of accounting practices.

Finally, we present a meta-heuristics-based MAO method for optimizing accounting methodologies. Meta-heuristics, known for their global optimization prowess, navigate complex search spaces to find optimal solutions. Integrating these algorithms with accounting methodologies allows for intelligent, self-adaptive parameter and strategy optimization during the accounting process. This adaptive approach enhances efficiency and enables automatic adjustment of accounting methods to meet varying enterprise requirements.

In summary, the intelligent accounting optimization method based on meta-heuristics and CNN proposed in this article realizes the intelligence and efficiency of accounting through three steps: multi-modal accounting feature extraction, accounting quality assessment and accounting method optimization. This method not only improves the accuracy of accounting but also provides high-quality financial data and decision-making suggestions for enterprises, which can help enterprises better respond to market changes and challenges.

## CNN-based multi-modal accounting feature extraction

We propose a CNN-based multi-modal accounting feature extraction method to deepen our understanding of accounting information and enhance method accuracy. Accounting data, comprising raw numbers, letters, and special symbols, is structured as "documents, vouchers, accounts, statements" during accounting events. These records capture objective observations and essential facts, utilizing distinct symbols to differentiate and identify various types of information. These symbols encompass quantitative numerical data and qualitative non-numerical data, forming the foundation of accounting information. Therefore, this study focuses on extracting multi-modal features from accounting information, encompassing commonly used documents, vouchers, and statements in accounting practices, as illustrated in Fig. 2.

In the accounting data processing workflow, accounting data primarily refers to the original data reflecting the increase and decrease in funds during the execution of

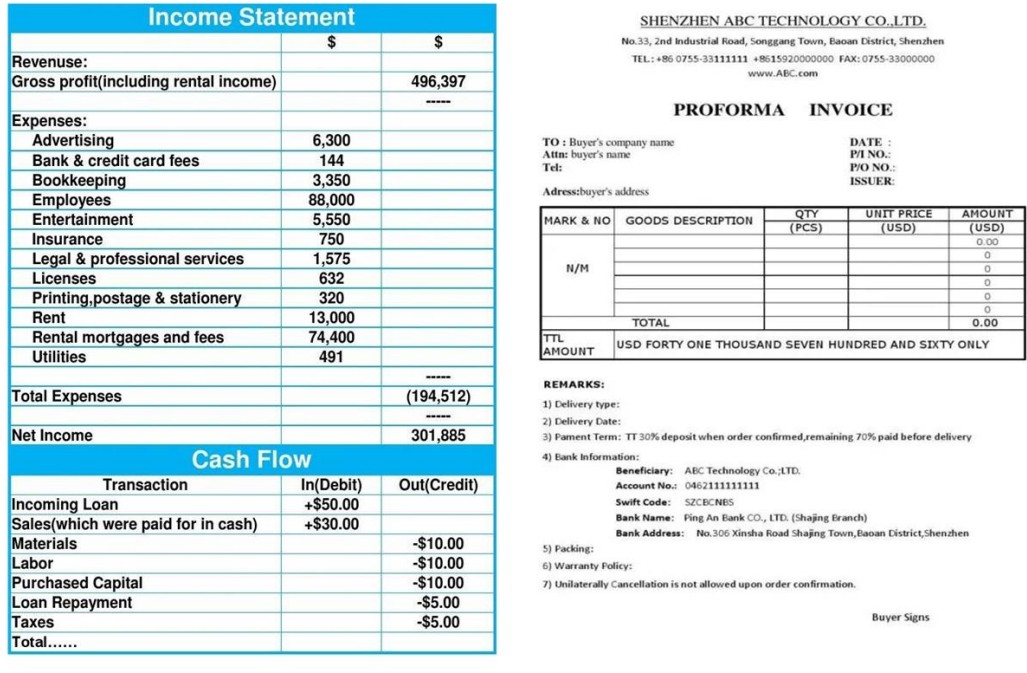

**Figure 2  Multi-modal accounting information.**

production and business activities or budget plans. It also covers a series of objective facts that do not directly involve changes in funds but must be recorded and reflected in accounting. These data originate from diverse sources and exhibit various types, characterized by their systematic, periodic, and continuous nature and their high reuse value. Given these unique properties, this article employs an optimized CNN technique to extract features from accounting data. Using CNN, valuable feature information is extracted from vast amounts of accounting data. These feature representations can reflect the complex structure and inherent relationships within accounting data, providing robust support for subsequent accounting analysis and forecasting. This article integrates a location enhancement module to explicitly model each textual element to enhance the even treatment of data sections and contextual backgrounds in current methods. The sequence of attention maps generated by the text detection recognizer is denoted by $h_{attn} \in \mathbb{R}^{T \times H \times W}$, where T is the maximum sequence length. Since the text length of different accounting images differs, the original attention map may be misleading due to attention drift. Therefore, this article adopts compression and expansion strategies to solve these problems. Denote the length of a character sequence as L, which $h_{attn}^{j}$ represents the attention map corresponding to the j-th text character. Since the text length varies from image to image, $\left\{ h_{attn}^{j} \right\}_{j=1}^{L}$ can be obtained by selecting the previous L effective attention map. Then, concatenate them, using a maximum operator to reduce the channel dimension

to 1. The result is denoted by $h_{score}$:

$$h_{score} = Max(Concat(h_{attn}^1, \ldots, h_{attn}^L)). \tag{1}$$

Use the C convolution kernel to upsample $h_{score}$, and use the Softmax function to obtain the result $h_{pos}$:

$$h_{pos} = Softmax(Conv(h_{score})). \tag{2}$$

The mean and variance contain the information of the image. To facilitate the subsequent registration between the image and the text, we perform instance normalization for $X_I$, to make the focus of the image more prominent. The normalized features are multiplied with $h_{pos}$ to obtain position-enhanced features $X_{pos}$:

$$X_{pos} = IN(X_I \bigotimes h_{pos}) \tag{3}$$

The key to the fusion of visual and semantic features is the registration of character regions rather than the attention to background, so feature selection technology is crucial. Instead of manual labelling and individual scoring methods, we can consider using easy-to-obtain attention maps for feature selection. Since pixel-wise character confidence is included in $h_{score}$, the one with the highest score in $h_{score}$ is selected, resulting in the foreground coordinate set F:

$$F := (m, n) : h_{score}(m, n) \in TopK(h_{score}). \tag{4}$$

Next, we use the coordinate set F as an index to collect the foreground character features in $X_{pos}$. To avoid loss of neighborhood information, a pixel-to-region strategy is used before collection, where each index pixel is representative of its neighbourhood by weighted summation of local regions:

$$X'_{pos} = \sum_{(\triangle m, \triangle n) \in N} \omega(\triangle m, \triangle n) X_{pos}(m + \triangle m, n + \triangle n). \tag{5}$$

where N represents the neighborhood displacement and $\omega(\cdot, \cdot)$ represents the weight of each displacement. Finally, the coordinate pair in F can index $X'_{pos}$ to obtain the selected feature $X_S = X'_{pos}(i, j)(i, j) \in F$.

When dealing with documents and vouchers in accounting information, we implement multimodal information extraction by embedding keyword fuzzy matching and the Levenshtein distance algorithm. After extracting information from different modalities, such as text, tables, and images, the system needs to integrate these pieces of information. During the integration process, it is crucial to ensure the accuracy and completeness of the information while also considering the correlations and logical relationships between different modalities. We further embed sequential reconstruction methods for information extraction for complex or specialised accounting documents and vouchers.

## Accounting quality assessment methods for accounting

The accounting quality evaluation method for accounting can further evaluate the accounting quality of enterprises by using the obtained features $X_S$, and provide standards for improving the accounting ability of enterprises.

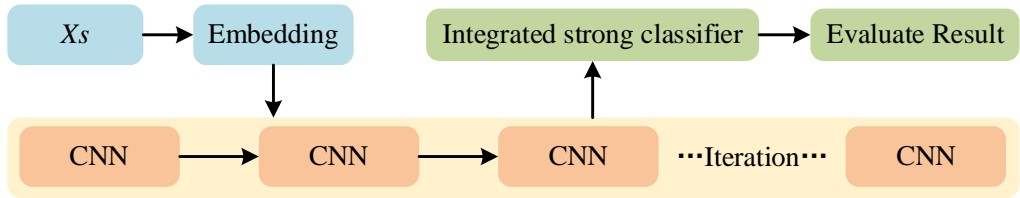

**Figure 3   Adaboost ensemble model based on CNN.**

To solve the problems of local minimization and over-fitting of multi-modal features $X_S$. We use the Adaboost ensemble learning model to optimize the evaluation results for the accounting objectives of enterprise accounting. The model performs the regression task by weighted combination of multiple weak learners. Firstly, a series of individual learners are generated, and then a suitable combination strategy is used to integrate them into a stronger learner, as shown in Fig. 3.

We employ a sequential structure to link multiple homogeneous weak classifiers, enhancing classifier stability and accuracy. The final learning outcome hinges on effectively combining the classification results from these weak classifiers. Central to our approach is the method for generating diverse weak classifiers. We achieve this by iteratively adjusting the sample weights and increasing focus on previously misclassified samples during subsequent training iterations. After completing all iterations, weights are assigned to each weak classifier based on its classification error rate, ensuring those with lower errors receive greater weights. Ultimately, these weighted weak classifiers are aggregated to form a robust learner with improved predictive accuracy.

If the data quality feature training set sample is $X$, and its corresponding quality class is $w$. The weight distribution for high-quality data is initialized, assigning equal weights $w_i$ to each training sample. Perform $m$ iterations, train each weak classifier in turn, select the weights of the weak classifier decision tree $B_m(x)$ to train the quality feature data, and calculate the error rate $e_m$ of the weak classifier at each iteration:

$$e_m = \frac{\sum_{N=1}^{Q} w_i^{(m)} |(c_i \neq B(x_i))}{\sum_{N=1}^{Q} w_i^{(m)}}. \tag{6}$$

Calculate the weights $\alpha_m$ for the weak classifier $B_m(x)$:

$$\alpha_m = \frac{1}{2} \ln \frac{1 - e_m}{e_m}. \tag{7}$$

Iterate until $m$ is equal to the set number of iterations $M$, and the final strong classifier $G(x)$ is obtained by weighting the sum of each weak classifier:

$$G(x) = \sin \left[ \sum_{m=1}^{M} \alpha_m B_m(x) \right]. \tag{8}$$

Adaboost can not only effectively deal with the local minimization problem caused by multi-modal features but also alleviate the over-fitting phenomenon to a certain extent, to

provide more accurate and stable evaluation results for the accounting goal of enterprise accounting $X_S$.

## Meta-heuristics-based accounting optimization method

Accounting optimization through meta-heuristics aims to enhance enterprise efficiency in accounting processes. These algorithms integrate meta-heuristic principles and optimization techniques to tackle the diverse and complex challenges inherent in enterprise accounting. This study optimises accounting methodologies by leveraging multi-modal features and quality assessment outcomes.

The approach employs meta-heuristic algorithms such as Genetic Algorithm (GA) and Simulated Annealing (SA) to solve accounting models and determine optimal enterprise operations. GA, renowned for addressing intricate optimization problems, operates with three key functions—selection, mutation, and crossover—each utilizing parameters to refine solutions. Such constraint is solved by maximizing the accounting quality assessment value, namely:

$$\max E = \sum_{i=1}^{n} c \times \left( \sqrt{(x_j + x_i)^2 + B^2} \right)^{n} \times D_L \tag{9}$$

$$\sum_{i=1}^{m} X_i = L \tag{10}$$

where L refers to the cost of the whole accounting process, each $x_i$ denotes the GA variable and $x_j$ represents the SA variable.

Mutation function is an important part of GA, and it includes Gaussian variograms, uniform variograms, and adaptive variograms. We chose to use a Gaussian variator with a shrinkage scale factor to successfully tune the minimum energy. When the shrinkage parameter is 1, the variance linearly decreases to 0 over the generations. This design makes the mutation process more flexible and controllable, which is helpful in boosting the performance of GA.

The SA model is locally based on the solution of a constrained optimization model. This constrained optimization model has the following steps:

(1) The corresponding objective $g_{j+1}$ is randomly generated by providing the starting temperature and selecting the variable $x_j$. The temperature then decreases as the number of cycles increases.

(2) Each time through the loop, a novel objective $g_{j+1}$ is created and compared with the current objective.

(3) If $g_{j+1} < g_j$, *replace g $_j$* with $g_{j+1}$ and enter the next cycle until the temperature value is really reduced.

The key of the SA model is to escape from the local optimal solution, which is determined by the Metropolis criterion. At high temperatures, the probability of which is wrong solution is high. As the temperature decreases, the probability of accepting the suboptimal solution is gradually reduced so the algorithm tends to search near the optimal solution.

The accounting optimization method based on Yuanqifa considers the diversity and complexity of enterprise accounting problems more comprehensively and is committed

**Table 1  Implementation parameters.**

| Parameters | Value |
|---|---|
| Initial learning rate | $9 \times 10^{-4}$ |
| Epoch | 600 |
| Batch-size | 60 |
| Decay | 0.92 |
| Gradient descent method | Adam |

to providing more effective accounting solutions. This method can be popularized and applied in different industries and enterprise types to promote the improvement of enterprise accounting efficiency.

# EXPERIMENT AND ANALYSIS

## Dataset and implement details

We tested intelligent accounting using a method combining CNN and meta-heuristics optimization, utilizing the Synthetic Financial Dataset available at Zenodo (DOI: 10.5281/zenodo.7543591).

Our training setup utilized hardware comprising an i5-14400F processor and 2 Nvidia RTX 4060Ti GPUs. Throughout the training phase, TensorFlow was our deep learning framework, configured with specific settings detailed in Table 1.

During model training, we incorporated a weight decay term with a value of 0.00009. These choices in test environments and configurations were made to ensure our method demonstrates superior efficiency and accuracy in addressing intelligent accounting challenges.

In this article, Accuracy (Acc) and Mean Average Precision (mAP) are employed as the criteria of intelligent accounting optimization method based on meta-heuristics and CNN, and the calculation formula is as follows:

$$Acc = \frac{Y(pr)}{Y(gt)} \tag{11}$$

$$mAP = \frac{\sum AP_i}{n} \tag{12}$$

where $pr$ and $gt$ refer to the predicted and true values, $n$ denotes the entire classes, and $AP_i$ is the average precision for the $i$ th class. AP is the area under the precision and recall curves and reflects the model's performance in each class. Therefore, the mAP means the average of the AP over all classes and is used to measure the average performance of the model over all classes. In addition, when calculating AP, we first need to get the precision and recall values.

## Ablation experiments

Table 2 presents detailed ablation experiments on intelligent accounting optimization methods using meta-heuristic algorithms and CNNs based on the Synthetic Financial

**Table 2  Ablation experiments.**

| MAFE | AQAA | MAO | mAP@0.5 | mAP@0.75 | mAP@0.95 | mAP@0.5:0.95 |
|------|------|-----|---------|----------|----------|--------------|
|      |      |     | 0.755   | 0.732    | 0.701    | 0.733        |
| O    |      |     | 0.779   | 0.752    | 0.739    | 0.753        |
|      | O    |     | 0.766   | 0.745    | 0.731    | 0.749        |
|      |      | O   | 0.786   | 0.754    | 0.748    | 0.767        |
| O    | O    |     | 0.799   | 0.779    | 0.768    | 0.785        |
| O    |      | O   | 0.812   | 0.801    | 0.783    | 0.803        |
|      | O    | O   | 0.824   | 0.809    | 0.798    | 0.815        |
| O    | O    | O   | 0.852   | 0.834    | 0.812    | 0.838        |

Dataset. We systematically evaluate the impact of three modules: MAFE, AQAA, and MAO, both individually and in various combinations, on model performance.

Firstly, individually introducing MAFE enhances mAP@0.5:0.95 to 0.739, AQAA contributes a score of 0.731 for mAP@0.5:0.95, and MAO achieves 0.748 for mAP@0.5:0.95. These results highlight the positive influence of each module in optimizing accounting processes.

Next, through detailed analysis in Figs. 4 and 5, we observe that combining these modules consistently improves model performance. Specific combinations demonstrate significant achievements, with some configurations reaching as high as 0.798 on mAP@0.95, underscoring the synergy between modules.

Finally, integrating all three modules—MAFE, AQAA, and MAO—yields compelling results: mAP@0.5 scores 0.852, mAP@0.75 scores 0.834, and mAP@0.95 scores 0.812. The overall mAP@0.5:0.95 reaches 0.838, affirming the effectiveness of our method and demonstrating the potential of multi-module combinations to enhance model performance in intelligent accounting optimization.

Our ablation experiments comprehensively explore the roles of MAFE, AQAA, and MAO modules in enhancing intelligent accounting optimization. These findings validate the efficacy of our approach and offer promising avenues for future research in this field.

## Compare our method and other methods

In our experiment on the Synthetic Financial Dataset, we evaluated the performance of the MAFE method by comparing it with DAMUN (*Feng et al., 2023*), FDGNet (*Zhang et al., 2023*), AdaMoW (*Zhang, Wu & Huang, 2023*), and Megcf (*Liu et al., 2023a*). The results, detailed in Fig. 6 and Table 3, demonstrate that MAFE outperforms all other methods across all evaluation metrics.

Specifically, MAFE achieved remarkable results across various accuracy metrics, with scores of 0.887 for Acc@1, 0.902 for Acc@5, 0.936 for Acc@10, and 0.966 for Acc@20. Compared to DAMUN, our method significantly improved 3.2% in Acc@1. Furthermore, MAFE surpassed FDGNet and AdaMoW in all indicators, outperforming them by 2.3% in Acc@5 and 1.4% in Acc@10, respectively. Lastly, MAFE achieved a 3.0% lead over Megcf in Acc@20. These comparisons underscore MAFE's superior performance in intelligent accounting. MAFE excels in accuracy, ranking first in Acc@1, and demonstrates significant

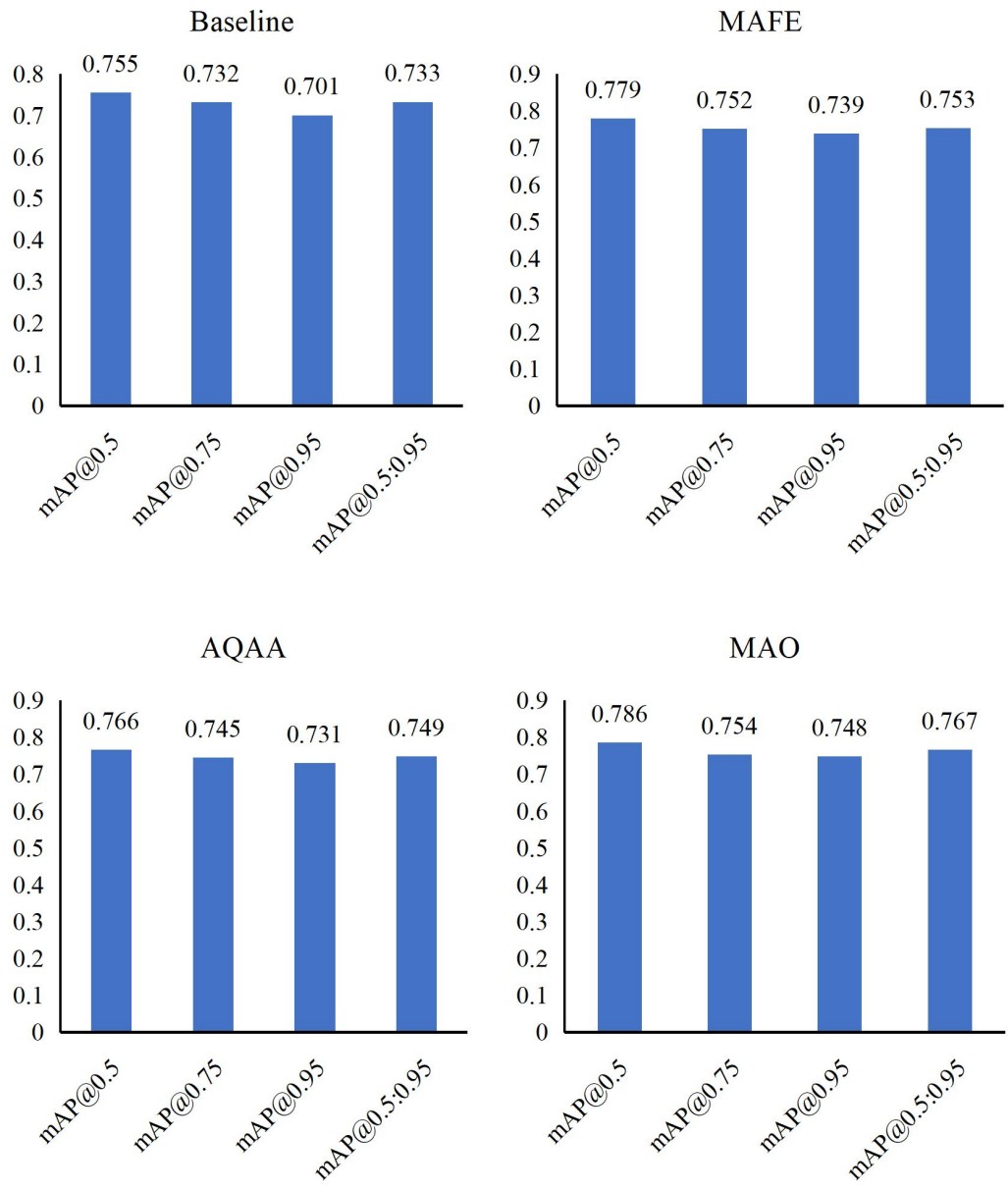

**Figure 4** Ablation experiments results in term of mAP with single module.

advancements in broader recommendation accuracy and scope. MAFE's superiority equips enterprises with more reliable and efficient intelligent solutions to address increasingly complex financial data and accounting challenges.

In our detailed analysis of AQAA's performance on the Synthetic Financial Dataset, we compared it against well-known methods CRFIQA (*Boutros et al., 2023*), PCQA (*Liu et al., 2023b*), and FAQA (*Idland et al., 2023*) to thoroughly assess its strengths and capabilities. As shown in Fig. 7 and Table 4, the results highlight AQAA's significant advantages across several key evaluation metrics.

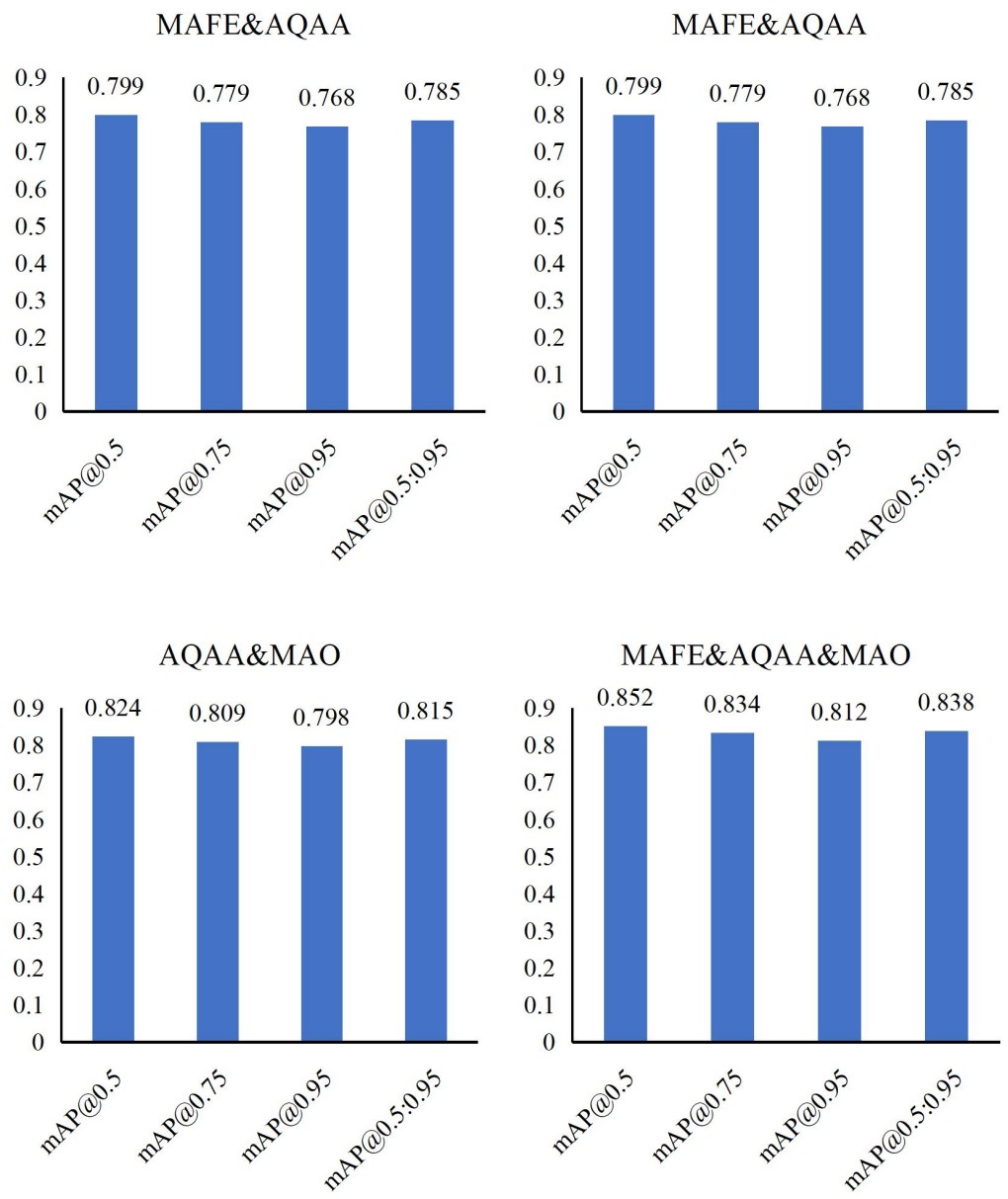

**Figure 5** Ablation experiments result in terms of mAP with different modules.

Compared to CRFIQA, AQAA excels in overall performance and leads by a significant 2.8% in the Acc@5 metric, exemplifying its prowess in precisely categorizing questions. Furthermore, when benchmarked against PCQA and FAQA, AQAA maintains its competitive edge. Specifically, AQAA outperforms PCQA with a remarkable 6.3% improvement in mAP@0.75, demonstrating its robustness in accurately identifying problematic areas. Additionally, AQAA surpasses FAQA with a 1.8% enhancement in Acc@5, further reinforcing its effectiveness in enhancing the accuracy of question identification. These findings collectively validate AQAA's prowess in improving the

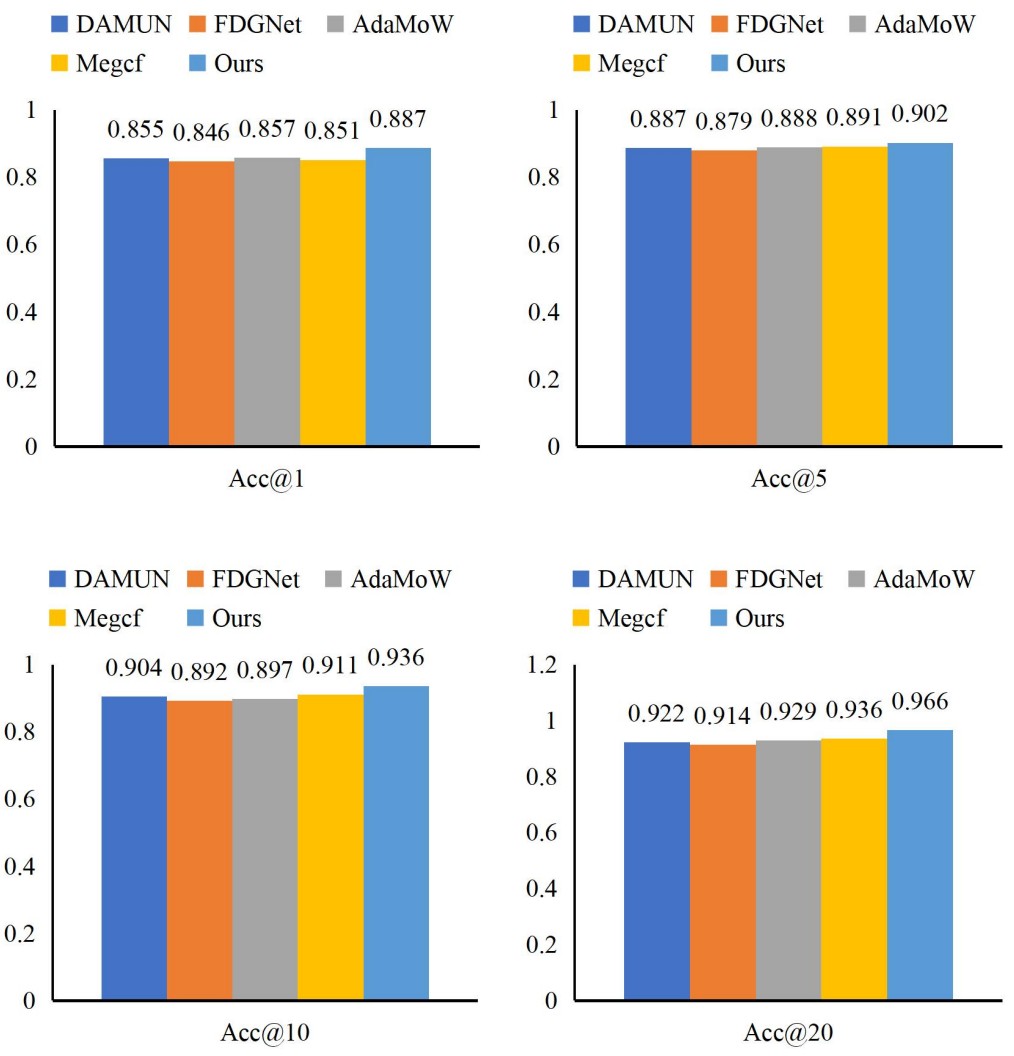

**Figure 6  Comparison of MAFE with other extraction methods.**

precision and effectiveness of question identification within the context of intelligent accounting optimization.

Through our detailed examination of MAO's performance on the Synthetic Financial Dataset, we have successfully enhanced AQAA's capabilities through parameter optimization. Comparing MAO with established benchmarks CRFIQA, PCQA, and FAQA, we observe that MAO achieves superior performance in Acc@1 and mAP@0.95 values, reaching 0.943 and 0.812, respectively. These detailed results are illustrated in Fig. 8.

MAO's optimization approach stands out among other advanced methods, exhibiting remarkable performance in accounting optimization. These superior metrics offer enterprises more efficient and dependable solutions for refining their accounting processes. The significant improvements in Acc@1 and mAP@0.95 values demonstrate MAO's robust adaptability in handling intricate financial data and accounting challenges. As a result,

**Table 3  Comparison of MAFE with other methods.**

| Methods | Acc@1 | Acc@5 | Acc@10 | Acc@20 |
|---|---|---|---|---|
| DAMUN | 0.855 | 0.887 | 0.904 | 0.922 |
| FDGNet | 0.846 | 0.879 | 0.892 | 0.914 |
| AdaMoW | 0.857 | 0.888 | 0.897 | 0.929 |
| Megcf | 0.851 | 0.891 | 0.911 | 0.936 |
| Ours | 0.887 | 0.902 | 0.936 | 0.966 |

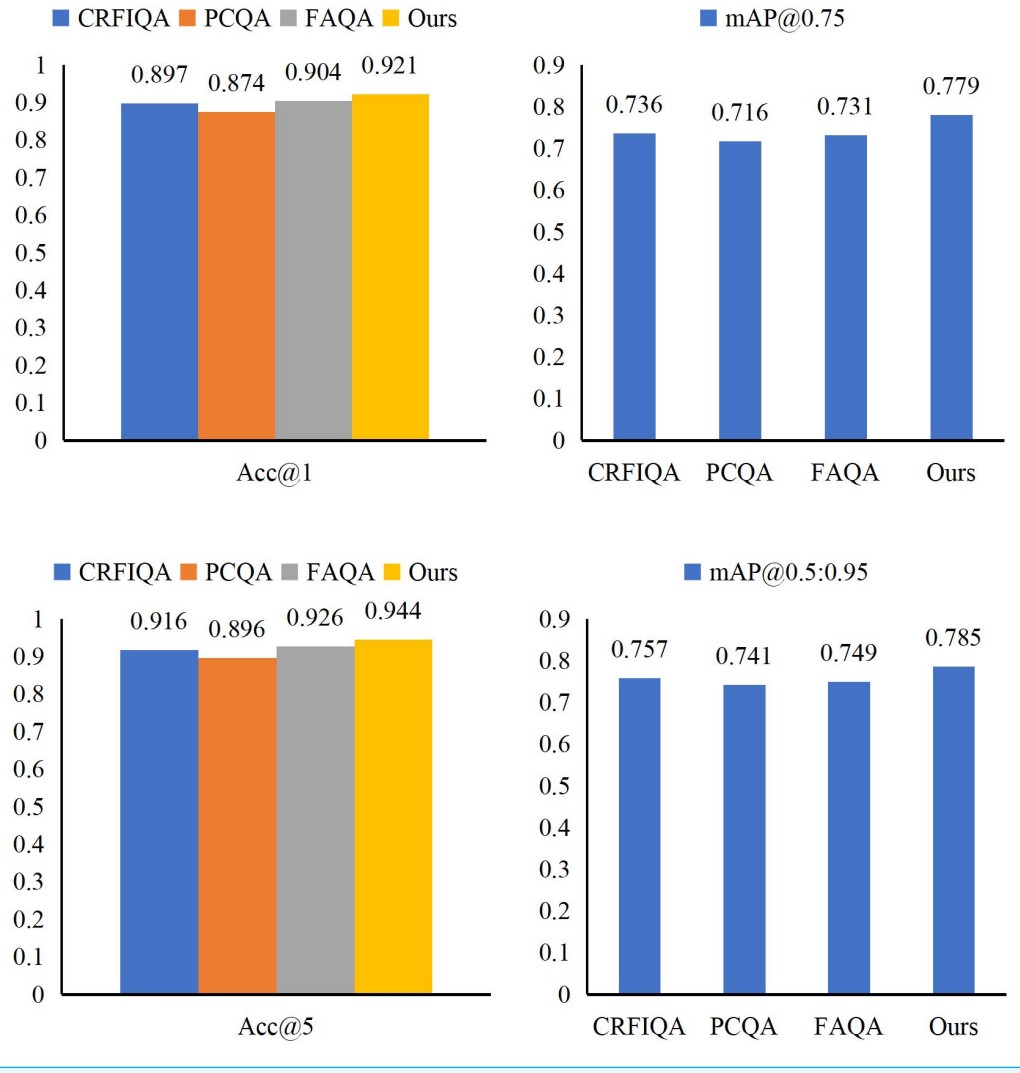

**Figure 7  The results of our AQAA.**

enterprises can confidently rely on MAO to provide flexible and effective accounting solutions, which are crucial for navigating dynamic economic environments and regulatory landscapes. MAO's method not only achieves substantial enhancements over existing benchmarks but also showcases its versatility in diverse domains and complex scenarios.

**Table 4  Comparison of AQAA with other methods.**

| Methods | Acc@1 | Acc@5 | mAP@0.75 | mAP@0.5:0.95 |
| --- | --- | --- | --- | --- |
| CRFIQA | 0.897 | 0.916 | 0.736 | 0.757 |
| PCQA | 0.874 | 0.896 | 0.716 | 0.741 |
| FAQA | 0.904 | 0.926 | 0.731 | 0.749 |
| Ours | 0.921 | 0.944 | 0.779 | 0.785 |

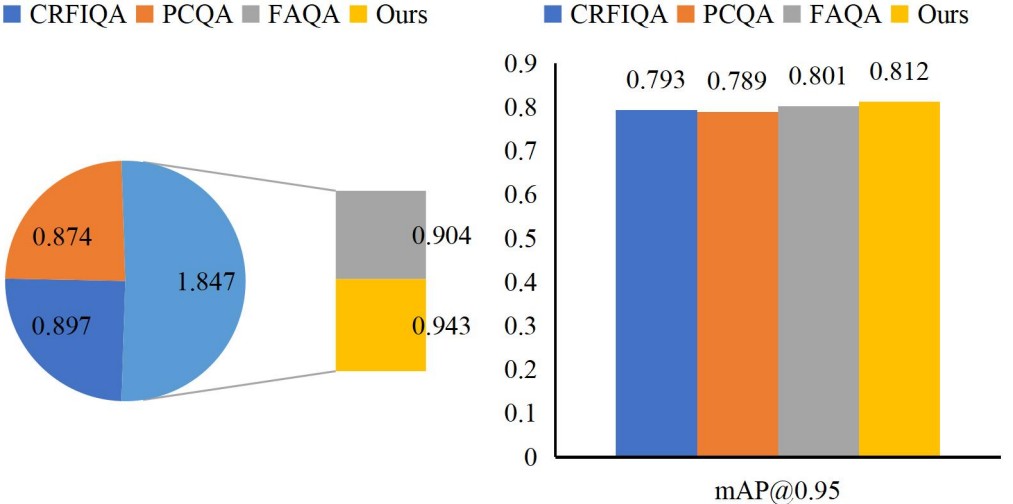

**Figure 8  Comparison of MAO with other methods.**

This comprehensive performance validates MAO's excellence in intelligent accounting, offering enterprises a reliable basis for decision-making and investors credible financial information.

Finally, to simulate the performance of our method in real-world scenarios, as shown in Fig. 9, we restructured the data to form tourism sample sets with varying numbers of real-world samples, including 20, 50, 100, 200, and 500 samples. By comparing the running time under different sample sets, we better understood our method's performance with different data scales. As shown in Fig. 7, we have recorded the computation time of MAFE, AQAA, and MAO under different sample sizes. Comparative analysis shows that regardless of the actual scenario, the system's computation time remains below 210 ms. This result indicates that our model can handle concurrent requests that may arise in real-world scenarios, thus providing users with stable and efficient services.

## Discussion

The experiments conducted on CNN-based multi-modal accounting feature extraction, accounting quality assessment, and meta-heuristics-based accounting optimization have yielded satisfactory results, validating the effectiveness of intelligent accounting optimization using meta-heuristics and CNN. This approach demonstrates exceptional

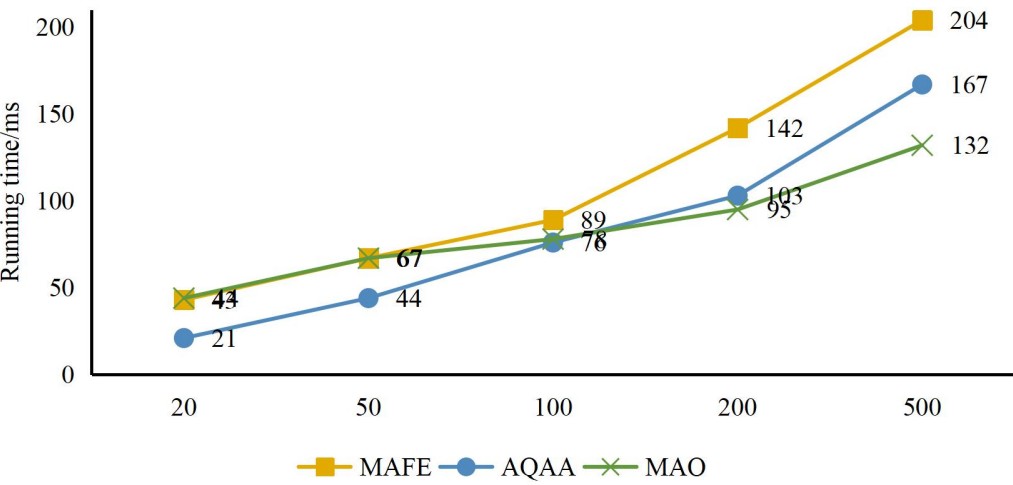

**Figure 9  System response efficiency test.**

performance across various environmental and data challenges, offering a comprehensive and practical solution for accounting tasks.

The CNN-based multi-modal accounting feature extraction technique successfully extracts crucial information from complex accounting data. By integrating data from diverse modalities, our method comprehensively reflects enterprise financial statuses and business outcomes, providing a robust foundation for subsequent accounting quality assessment and optimization. Secondly, our adopted method for accounting quality assessment incorporates deep data mining and machine learning algorithms, enhancing the objectivity and accuracy of evaluation results beyond traditional financial indicators. This capability aids enterprises in the timely identification of risks and issues, facilitating informed decision-making. Lastly, the meta-heuristics-based accounting optimization method optimizes accounting processes by intelligently adjusting model parameters and structures. This not only boosts accounting accuracy and efficiency but also reduces manual intervention, thereby easing the workload on financial personnel.

In summary, our intelligent accounting optimization method leveraging meta-heuristics and CNN proves to be an efficient, reliable solution with broad application potential. We aim to explore additional optimization strategies and technological advancements to enhance accounting accuracy further and support enterprise development.

## CONCLUSION

We introduce an innovative intelligent accounting optimization method combining metaheuristic algorithms and CNN to enhance enterprise accounting capabilities. Initially, we employ a CNN-based multi-modal accounting feature extraction approach to derive multi-modal representations from accounting data, encompassing financial documents, vouchers, and related information. Building upon this foundation, we develop an accounting quality evaluation method and establish a comprehensive accounting evaluation

system. Leveraging these multi-modal features and the evaluation system, we integrate a meta-heuristic-based accounting optimization method to enhance accounting processes intelligently. Experimental results demonstrate significant achievements, including an accuracy of 0.943 and an mAP score of 0.812. These outcomes validate our method as an effective technical solution for elevating accounting quality. In the future, we will continue to deepen and expand this intelligent accounting optimization approach. First, we plan to optimize the CNN model further to extract more refined and comprehensive accounting data features. By introducing more advanced network structures and algorithms, we can capture more subtle financial changes, thereby enhancing the accuracy and efficiency of accounting analysis. Moreover, we will explore how to integrate meta-heuristic algorithms with CNN better to achieve more intelligent and efficient accounting optimization.

### Funding
The author received no funding for this work.

### Competing Interests
The author declares there are no competing interests.

### Author Contributions
- Yanrui Dong conceived and designed the experiments, performed the experiments, analyzed the data, performed the computation work, prepared figures and/or tables, authored or reviewed drafts of the article, and approved the final draft.

### Data Availability
The code is available in the Supplementary File.

The dataset is available at Zenodo: Mavsar, M. (2022). Video-Trajectory Robot Dataset [Data set]. Zenodo. Available at https://doi.org/10.5281/zenodo.6337847.

### Supplemental Information
Supplemental information for this article can be found online at http://dx.doi.org/10.7717/peerj-cs.2281#supplemental-information.

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
