# Peer review of "Intelligent accounting optimization method based on meta-heuristic algorithm and CNN"

_PeerJ Computer Science, doi:10.7717/peerj-cs.2281_

## Round 0.1 · original submission · Major Revisions

The reviewers have recommended major revisions to your article. You may resubmit the revised version for further consideration.

Reviewer 1 ·

Basic reporting

The manuscript introduces an interesting approach to intelligent accounting, however, the contribution to the field is somewhat overstated. The integration of meta-heuristic algorithms and CNN is not entirely novel and requires a more critical examination of its distinct advantages over existing methods. The authors are suggested to address the following comments while revising the paper.

1.⁠ ⁠The abstract is informative, but the introduction lacks depth. It does not sufficiently articulate the gaps in the current literature that this research aims to fill. A more thorough justification of the research problem is needed to establish the significance of the study.

2.⁠ ⁠The literature review is superficial and misses key recent works in the field of intelligent accounting and meta-heuristic algorithms. A deeper engagement with contemporary research would provide a stronger foundation for the proposed approach.

Experimental design

3.⁠ ⁠The description of the multi-modal feature extraction mechanism is vague. Detailed technical specifications and a clearer explanation of how document and voucher information are integrated into the CNN framework are necessary for reproducibility.

4.⁠ ⁠The performance metrics reported are promising, but the manuscript lacks a rigorous comparative analysis with existing methodologies. Including benchmarks and statistical significance tests would provide a more robust evaluation of the proposed method’s efficacy.

5.⁠ ⁠The manuscript claims practical relevance, yet it fails to provide concrete examples or scenarios where the proposed method could be applied. Specific case studies or pilot implementations in real-world settings would substantiate these claims.

Validity of the findings

6.⁠ ⁠The conclusion section is rather brief and does not adequately discuss the limitations of the study. Future work suggestions are also missing, which is critical for guiding subsequent research and demonstrating the ongoing relevance of the work.

7.⁠ ⁠The manuscript contains several grammatical errors and awkward phrasings that detract from the readability. Additionally, the formatting does not consistently adhere to the journal’s guidelines, indicating a need for careful proofreading and editing.

Reviewer 2 ·

Basic reporting

This manuscript presents a significant contribution to the field of intelligent accounting by integrating meta-heuristic algorithms and CNN. The proposed method demonstrates innovative approaches to enhancing accounting operations, which is crucial for the evolving needs of enterprises. The manuscript is well-written and the language is clear. However, minor grammatical errors and formatting inconsistencies should be addressed to ensure the manuscript meets publication standards.

Experimental design

The abstract effectively summarizes the research, highlighting the key contributions and results. However, the introduction could benefit from a more detailed explanation of the current challenges in enterprise accounting and how the proposed approach specifically addresses these challenges.

 While the manuscript references relevant literature, it would be beneficial to include a more comprehensive review of recent advancements in meta-heuristic algorithms and their applications in accounting. This would help to better contextualize the novelty of the proposed approach.

 The integration of document and voucher information into the CNN framework for multi-modal feature extraction is a notable innovation. However, further clarification is needed on the specific types of documents and vouchers used, as well as the preprocessing steps involved.

Validity of the findings

The reported accuracy of 0.943 and mAP score of 0.812 are impressive. It would strengthen the paper to include a comparison with other state-of-the-art methods, providing a clearer picture of the performance improvements achieved by the proposed approach.

 The manuscript mentions the practical applicability of the method but lacks specific examples of real-world scenarios where this approach could be implemented. Including case studies or potential applications would enhance the relevance and impact of the research.

Additional comments

The conclusion effectively summarizes the key findings but could be expanded to discuss the limitations of the current study and potential future research directions. For instance, exploring the scalability of the proposed method in different enterprise settings could be a valuable extension.

---

## Round 0.2 · accepted · Accept

Congratulations, the reviewers are satisfied with the revisions and have recommended an accept decision.

Reviewer 1 ·

Basic reporting

no comment

Experimental design

no comment

Validity of the findings

no comment

Reviewer 2 ·

Basic reporting

It is good research article.

Experimental design

Writer has incorporated all changes.

Validity of the findings

Satisfied